# Prognostic Values of Tissue and Serum Angiogenic Growth Factors Depend on the Phenotypic Subtypes of Colorectal Cancer

**DOI:** 10.3390/cancers15153871

**Published:** 2023-07-29

**Authors:** Jussi Herman Kasurinen, Jaana Hagström, Tuomas Kaprio, Sirpa Jalkanen, Marko Salmi, Camilla Böckelman, Caj Haglund

**Affiliations:** 1Translational Cancer Medicine Research Program, Faculty of Medicine, University of Helsinki, 00100 Helsinki, Finlandcamilla.bockelman@helsinki.fi (C.B.); caj.haglund@hus.fi (C.H.); 2Department of Pathology, University of Helsinki and Helsinki University Hospital, 00100 Helsinki, Finland; 3Department of Oral Pathology and Radiology, University of Turku, 20014 Turku, Finland; 4Department of Surgery, University of Helsinki and Helsinki University Hospital, 00100 Helsinki, Finland; 5MediCity Research Laboratory and Institute of Biomedicine, University of Turku, 20014 Turku, Finland

**Keywords:** colorectal cancer, colon cancer, immunohistochemistry, CMS, tumor phenotype, VEGF, bFGF, PDGF-bb, survival

## Abstract

**Simple Summary:**

In this study, we showed that the prognostic value of serum concentrations and the tissue expressions of angiogenic growth factors VEGF, bFGF, and PDGF-bb vary according to the phenotypic subtype of the tumor. Patients were classified into phenotypic subgroups (*immune*, *canonical*, *metabolic*, and *mesenchymal*). Preoperative serum concentrations and tissue expressions of VEGF, bFGF, and PDGF-bb were determined among each phenotypic subgroup. A high tissue expression and serum concentration of angiogenic growth factors seem to indicate improved prognosis among the *metabolic* subgroup. Among *immune* patients, a high VEGF serum concentration is associated with worse prognosis. Moreover, a high serum concentration of bFGF indicated improved prognosis among *canonical* patients.

**Abstract:**

We classified colorectal cancer (CRC) patients into four phenotypic subgroups and investigated the prognostic value of angiogenic growth factors across subgroups. Preoperative serum concentrations and tissue expressions of VEGF, bFGF, and PDGF-bb were determined among 322 CRC patients. We classified patients into phenotypic subgroups (*immune*, *canonical*, *metabolic*, and *mesenchymal*) according to a method described in our earlier work. Among the *metabolic* subgroup, patients with high serum concentrations of VEGF, bFGF, or PDGF-bb exhibited a significantly improved prognosis. Moreover, those with high VEGF tissue expressions exhibited a significantly improved prognosis among patients in the *metabolic* subgroup. Among *immune* patients, a high VEGF serum expression is associated with a worse prognosis. A high serum bFGF concentration is associated with a favorable prognostic factor among patients with a *canonical* tumor phenotype. A high PDGF-bb tissue expression is associated with non-metastasized disease and with the *immune*, *canonical*, and *metabolic* subtypes. To our knowledge, this is the first study to show that the prognostic value of angiogenic growth factors differs between phenotypic subtypes.

## 1. Introduction

The incidence of colorectal cancer (CRC) is the third highest of all malignancies, with over 1,800,000 cases occurring each year, and the second-most common cause of cancer death, resulting in over 800,000 patient deaths annually [1,2]. The molecular diversity of colorectal tumors is thought to partly explain the differences in clinical outcomes.

Angiogenic factors, including the vascular endothelial growth factor (VEGF), basic fibroblast growth factor (bFGF), and platelet-derived growth factor (PDGF-bb), orchestrate blood vessel formation during the embryogenic and postnatal development of healthy tissues. In addition, angiogenic factors are also essential in the formation of the tumor neovasculature, allowing tumor growth and progression [3,4].

Endothelial cells express two receptors, VEGFR1 and VEGFR2, that bind VEGF, the key mediator of angiogenesis in cancer and the primary target of anti-angiogenic therapies [5,6]. A high tissue expression and serum concentration of VEGF appear indicative of a poor prognosis in CRC [7,8,9]. While elevated serum VEGF levels seem to indicate a worse prognosis in CRC, conflicting results regarding its diagnostic value have been reported [10,11]. A variety of fibroblast growth factor (FGF) receptors bind bFGF (or FGF2), which prompts a spectrum of activities related to cell survival and proliferation. In addition, FGF plays an important role in the regulation of endothelial cell differentiation, migration, and proliferation [12]. A high serum bFGF concentration and bFGF tissue expression are associated with advanced CRC disease [13,14,15]. PDGF-bb is a dimeric growth factor with four other known isoforms. It binds to two receptors, PDGFRA and PDGFRB [16]. PDGF-bb is known to promote pericyte retention and non-aberrant angiogenesis [17]. In CRC, circulating PDGF-bb is elevated compared with benign neoplasia. Additionally, the PDGF-bb serum concentration has been shown to be higher in advanced disease [18]. Finally, an increased PDGF-bb tissue expression indicates poor survival among CRC patients [19].

The Colorectal Cancer Subtyping Consortium, using six independent gene expression-based CRC classification systems [20,21,22,23,24,25,26], classifies the disease into four consensus molecular subtypes (CMS). *Immune* (CMS1) tumors are characterized by a microsatellite instability and a high amount of tumor-infiltrating lymphocytes (TILs) [26,27]. *Canonical* (CMS2) tumors have induced the canonical Wnt pathway and are highly proliferative [28]. Metabolically mutated *metabolic* (CMS3) tumors have undergone changes to a glycolytic state and can better survive in a hostile environment [29]. Genes implicated in the epithelial–mesenchymal transition are upregulated in *mesenchymal* (CMS4) tumors, which show a high infiltration of stromal cells [26].

The gene expression-based analysis of clinical tumor samples remains time-consuming and expensive. However, based on the CMS classification, Roseweir et al. [30] presented another method of dividing tumors into subgroups according to phenotypic features using immunohistochemistry. CD3 and CD8 immunostainings were used to recognize the *immune* phenotypic subgroup (representing CMS1) characterized by a high infiltration of TILs. The proliferative *canonical* tumors (representing CMS2) were recognized by assessing the proliferation index. A high stromal infiltration of malignant cells, assessed using the tumor stroma percentage (TSP), was used to identify the *mesenchymal* phenotypic subgroup (representing CMS4). The remaining tumors with a low infiltration of TILs, a low proliferation index, and a low TSP represented the *metabolic* subgroup (representing CMS3). In our earlier work, we showed that immunohistochemically determined phenotypic subtypes of CRC predicted patient outcomes in a manner resembling the transcriptome-based CMS classification. Moreover, the *immune* subtype was associated with right-sided tumors and female gender, and the *mesenchymal* subtype was associated with advanced disease resembling the gene expression-based CMS classification [31].

Since the genomic alterations and tumor pathways differ between the CMS subgroups, drug responses and the tumor biomarker expression might also differ. An earlier study indicated that leucovorin–5-flurouracil–oxaliplatin (FOLFOX) treatment appears superior to capecitabine–oxaliplatin (CAPOX) treatment among *immune* patients, while the two treatment options showed similar responses in the remaining three phenotypic subgroups [30].

To our knowledge, no previous studies have investigated the prognostic role of angiogenic growth factors among different CRC phenotypes or CMS subgroups, although angiogenic growth factors in CRC have been studied extensively. In this study, we divided CRC patients into four phenotypic subgroups using immunohistochemistry, and assessed the prognostic role of tissue expressions and serum concentrations of VEGF, bFGF, and PDGF-bb among each subgroup.

## 2. Methods

### 2.1. Patients

We included 322 surgically treated CRC patients in this retrospective study. Patients were treated between 1998 and 2003 with a curative intent in the Department of Surgery at Helsinki University Hospital, Finland. The median patient age was 66.9 years [interquartile range (IQR) 57.5–75.7]. Among patients, 48.1% were female. The follow-up period was from the time of surgery until 2020. The median follow-up time was 6.39 years (IQR 2.11–16.2). At the end of follow-up, 38.2% of patients were living, and 28.5% had died due to CRC. Tumors were divided by location: right-sided colon tumors, left-sided colon tumors, and rectal tumors. Tumors located proximal to the splenic flexure were classified as right-sided tumors, while those located in the splenic flexure were excluded from the analysis of tumor location. Patients’ clinicopathological features are summarized in Appendix A.

Essential clinical information was extracted from patients’ medical records from the hospital database. Survival and cause-of-death information were provided by the Population Register Center of Finland and Statistics Finland. The study protocol was approved by the Surgical Ethics Committee of Helsinki University Hospital (Dnro HUS 226/E6/06, extension TMK02 §66 17.4.2013). The National Supervisory Authority of Health and Welfare granted permission to study the archived tissue samples without requiring individual consent (Valvira Dnro 10041/06.01.03.01/2012).

### 2.2. Serum Markers

Preoperative blood samples were collected for 320 patients and stored at −80 °C until assayed. The Bio-Plex Pro™ Human Cytokine 27-plex Assay (#M500KCAF0Y) and the 21-plex Assay (#MF0005KMII; Bio-Rad, Hercules, CA, USA) were used to simultaneously analyze 48 different cytokine concentrations, including VEGF, bFGF, and PDGF-bb. Only the VEGF, bFGF, and PDGF concentrations were used for the purposes of this study. Previously, we investigated the prognostic values of the other 45 cytokines in CRC [32].

### 2.3. Preparation of Tissue Samples

Formalin-fixed paraffin-embedded tumor samples were retrieved from the archives of the Department of Pathology at Helsinki University Hospital. Here, 1 mm-diameter tissue cores were punched out from representative areas of the tumor cores, which were earlier marked on HE slides by an experienced pathologist (JH). The 1 mm tissue cores were then embedded into tissue microarray (TMA) paraffin blocks using a semiautomatic tissue arrayer (Beecher Instruments Inc., Silver Spring, MD, USA). Each TMA block containing up to 56 tumor tissue cores was subsequently cut into 4 µm sections to enable immunohistochemical staining.

### 2.4. Immunohistochemistry

Our previous work described the immunohistochemical stainings of CD3, CD8, Ki67, and cytokeratin used to determine the phenotypic subgroups [31].

The immunohistochemistry of the angiogenic growth factors VEGF, bFGF, and PDGF-bb was carried out using a similar method used to describe Ki67 and cytokeratin stainings [31]. First, slides were pretreated for 15 min with an EnVision Flex target retrieval solution (Dako, Santa Clara, CA, USA, DM828) at 98 °C in a pretreatment module (Agilent Technologies Inc., Dako, Santa Clara, CA, USA). Next, the pretreated slides were incubated with primary antibodies (mouse polyclonal anti-human VEGF (PharMingen, San Diego, CA, USA, diluted to 1:100), rabbit polyclonal anti-human bFGF (Bioss, Boston, Massachusetts, USA, diluted to 1:800), and rabbit polyclonal anti-human PDGF-bb (Thermo Scientific, Waltham, Massachusetts, USA, diluted to 1:500) in an Autostainer 480 (Lab Vision Corp, Fermont, CA, USA) overnight at room temperature. Subsequently, the slides were treated for 20 min with HRP-labeled EnVision Flex/HRP secondary antibodies (Dako, Santa Clara, CA, USA, SM802), which were then visualized by 10 min incubation with EnVision Flex DAB chromogen (Dako, Santa Clara, CA, USA, DM827). Finally, the slides were counterstained with Meyer’s hematoxylin and washed in tap water.

### 2.5. Scoring of Samples

All samples were independently scored by two investigators (JK and JH) who were blinded to the clinical data. All scoring results were reviewed, and disagreements about scores were discussed until consensus was achieved. The scoring of CD3, CD8, Ki67, and cytokeratin stainings and the determination of the phenotypic subgroups are explained in detail in our previous work [31]. Briefly, tumors with a high infiltration of CD3- and CD8-positive lymphocytes were classified as the *immune* subgroup. Of the remaining samples, those with high TSP (determined by cytokeratin staining) were allocated to the *mesenchymal* subgroup. The proliferation index (determined by Ki67 staining) was used to assign the remaining tumor samples into *canonical* and *metabolic* subgroups so that those with a high proliferation index were classified as *canonical.* The distribution of patients according to phenotypic subgroups is summarized in Appendix A.

The staining intensities of VEGF, bFGF, and PDGF-bb in the cytoplasm of tumor cells were scored on a scale from 0 to 3 (0, negative staining intensity; 1, weak staining intensity; 2, moderate staining intensity; and 3, strong staining intensity). The staining pattern of the tumor cells was even in each TMA spot. Representative images of the immunohistochemical staining categories appear in Figure 1.

For the survival analysis, the serum concentrations of the angiogenic growth factors were dichotomized as low and high. The median values of the VEGF, bFGF, and PDGF-bb serum concentrations were determined for each phenotypic subgroup and used as the cut-off values. The continuous VEGF, bFGF, and PDGF-bb serum concentration values were used for the analysis of correlations. For the tissue expressions of VEGF, bFGF, and PDGF-bb, samples with negative and weak staining intensities were considered low and samples with moderate and high staining intensities were considered high.

### 2.6. Statistical Analyses

All statistical analyses were performed using IBM SPSS Statistics 27 for Mac (IBM SPSS Statistics, version 27; SPSS, Inc., Chicago, IL, USA, an IBM Company). The associations and correlations between the angiogenic growth factors, phenotypic subtypes, and relevant clinicopathological variables were evaluated using Pearson’s chi-square test and Spearman’s rank correlation. Survival curves were constructed using the Kaplan–Meier method and compared using the log-rank test. The Cox proportional hazard model was used to calculate the hazard ratios (HRs). We used a two-tailed threshold of statistical significance set to *p* < 0.05 in all analyses.

## 3. Results

### 3.1. Correlations between Clinicopathological Variables and Serum Angiogenic Growth Factor Concentrations

Table 1 summarizes the correlations between the VEGF, bFGF, and PDGF-bb serum concentrations and clinicopathological variables. We found no significant correlations between the VEGF or bFGF serum concentrations and clinicopathological variables. However, the PDGF-bb serum concentration negatively correlated with age (r_s_ = 0.309, *p* < 0.001, Table 1). We also found no significant correlations between the tissue expressions and serum concentrations of VEGF, bFGF, and PDGF-bb (Appendix A).

### 3.2. Associations between Clinicopathological Variables and Tissue Expressions of Angiogenic Growth Factors

Table 2 summarizes the associations between the VEGF, bFGF and PDGF-bb tissue expressions and clinicopathological variables. A high VEGF expression is associated with the *immune*, *canonical*, and *metabolic* subtypes (*p* < 0.001, Table 2). A high bFGF expression is associated with the *canonical* subtype (*p* = 0.022) and rectal tumors. Finally, a high PDGF-bb expression is associated with the *immune*, *canonical*, and *metabolic* subtypes (*p* < 0.001, Table 2).

### 3.3. Survival Analysis

Among the *immune* subtype, we found a 5-year disease-specific survival (DSS) for patients with a high VEGF serum concentration of 80.2% [95% confidence interval (CI) 62.8–97.6%], which increased to 90.6% (95% CI 80.4–100%) for patients with a low VEGF serum concentration (log-rank test: *p* = 0.038, Figure 2a). None of the patients with the *metabolic* phenotype and a high VEGF serum concentration died due to CRC. Patients with the *metabolic* tumor phenotype and a low VEGF serum concentration had a 5-year DSS of 58.8% (95% CI 38.0–79.6%, log-rank test: *p* = 0.024, Figure 2c). Serum VEGF was not a significant prognostic factor in patients with *canonical* or *mesenchymal* tumors (Figure 2b,d).

Among the metabolic subgroup, 5-year DSS was 77.7% (95% CI 60.3–95.1%) for patients with a high tissue expression of VEGF and 42.3% (95% CI 5.65–79.0% log-rank test: *p* = 0.012, Figure 3c) for patients with a low VEGF expression. Among the *immune*, *canonical*, and *mesenchymal* subtype patients, VEGF tissue expression was not a significant prognostic factor (Figure 3a,b,d).

A high serum bFGF concentration was a favorable prognostic factor among patients with *canonical* (HR 3.88, 95% CI 1.06–14.2, log-rank test: *p* = 0.027, Appendix A) and *metabolic* (HR 5.31, 95% CI 1.15–24.6, log-rank test: *p* = 0.017, Appendix A) tumor phenotypes. Among all other subgroups, the bFGF tissue expression was not a significant prognostic factor.

Furthermore, patients with the *metabolic* tumor phenotype and a high PDGF-bb serum concentration exhibited a better prognosis (HR 3.83, 95% CI 1.05–14.6, log-rank test: *p* = 0.035, Appendix A). PDGF-bb tissue expression was not a significant prognostic factor among any of the subgroups.

## 4. Discussion

In this study, we clarified the prognostic value of angiogenic growth factors VEGF, bFGF, and PDGF-bb, which varied according to the phenotypic subgroups in CRC. Specifically, a high VEGF serum concentration and tissue expression emerged as favorable prognostic factors among patients with the *metabolic* tumor phenotype. Moreover, patients with the *metabolic* tumor phenotype and high serum concentrations of bFGF and PDGF-bb exhibited a better prognosis. Intriguingly, patients with the *immune* tumor phenotype and a high serum concentration of VEGF exhibited a worse prognosis.

Earlier studies indicated that a high serum concentration of VEGF is associated with advanced disease and indicated a worse prognosis in CRC patients [9,10,11,12,13,14,15,16,17,18,19,20,21,22,23,24,25,26,27,28,29,30,31,32,33]. In the current study, we found that the prognostic effect of the serum VEGF concentration differed between phenotypes. Interestingly, in our cohort, none of the *metabolic* subgroup patients with a high serum VEGF concentration died during follow-up and a high serum VEGF indicated an improved prognosis in this subgroup. Furthermore, similar to earlier studies, patients with a high VEGF concentration and the *immune* subtype tumor exhibited a worse 5-year DSS.

Others have suggested that a high VEGF expression in the tumor tissue leads to abundant tumor angiogenesis and cancer progression [3]. Similar to the serum VEGF concentration, a high VEGF tissue expression determined using immunohistochemistry [7] and an mRNA expression analysis [34] correlated with an advanced stage of disease and a worse survival in earlier CRC studies. However, our results indicated that *metabolic* subtype patients with a high VEGF tissue expression experienced an improved prognosis.

In addition to VEGF, bFGF and PDGF-bb are relevant components in the induction of tumor angiogenesis. Earlier studies suggested that patients with high bFGF serum concentrations [13] and tissue expressions [14] experienced a worse prognosis in CRC. Correspondingly, previous research indicated that a high PDGF-bb serum concentration [35] is associated with advanced disease and patients with a high tissue expression of PDGF-bb exhibited a worse CRC prognosis [19]. However, our data illustrated that *metabolic* phenotype patients with a high PDGF-bb tissue expression or bFGF serum concentration exhibited a significantly better prognosis. Moreover, PDGF-bb and bFGF were not significant prognostic factors among any of the other phenotypic subgroups. Thus, it seems that angiogenic growth factor levels may carry contradictory impacts on outcomes among *metabolic* phenotype patients, indicating that the subgrouping of CRC tumors is useful.

To our knowledge, no previous studies investigated VEGF, bFGF or PDGF-bb associations by categorizing tumors according to molecular subtypes or phenotypic subgroups. Our analysis implies that a high tissue expression of all angiogenic growth factors studied is associated with the *canonical* tumor phenotype. *Canonical* tumors have genetic alterations inducing the canonical Wnt pathway [26]. Earlier studies indicated that an increased Wnt2 acts as a pro-angiogenic growth factor and might explain the elevated expression of angiogenic growth factors in *canonical* tumors [36].

According to Guinney et al. [26], metabolically mutated CMS3 tumor cells have undergone mutations to a glycolytic state and can withstand hypoxia better. Consequently, abundant angiogenesis leading to a more oxygen-rich tumor microenvironment might be disadvantageous for CMS3 or *metabolic* tumor cells. Antitumoral components of the tumor microenvironment, such as CD3 and CD8 immune cells [37], might function better in an oxygen-rich environment, explaining the improved prognosis of *metabolic* phenotype patients with the high levels of serum and tissue angiogenic growth factors observed in this study. Another explanation for this result might be that *metabolic* tumors are infiltrated with a large amount of VEGF expressing myeloid-derived cells, namely, macrophages. Stockmann et al. [38] demonstrated that the loss of myeloid-derived VEGF accelerates, rather than decelerates, tumor growth and progression.

Anti-VEGF therapy is often recommended as a first-line oncological treatment in metastatic CRC [39], with a high tumor VEGF expression predicting a better response to it [40]. Moreover, an earlier study indicated that CMS3 patients with metastatic CRC responded better to leucovorin–5-flurouracil–irinotecan (FOLFIRI) + cetuximab therapy compared with FOLFIRI + Bevacizumab therapy [41]. In this study, we found that patients with high levels of angiogenic growth factors exhibited a better prognosis among the *metabolic* subgroup. Consequently, the benefit of suppressing tumor angiogenesis might vary according to the tumor phenotype or the molecular subgroup. However, to confirm these conclusions, additional research is needed.

The strengths of this study include the long follow-up time allowing us to precisely determine patient outcomes, a well-characterized cohort, and reliable survival data. However, the single-center setting, the lack of information regarding adjuvant or neoadjuvant therapies, and incomplete data on co-morbidities reflect several limitations to this study. Moreover, although the total number of patients in this study is reasonable, once divided into phenotypic subgroups, the number of patients within subgroups decreases. Thus, the statistical findings must be interpreted with caution. Further studies with preferably larger cohorts are needed to confirm our findings reported here.

In addition, compared with genomic assessments, the immunohistochemical methods used in this study are accessible at a low cost and are easily translated to clinical practice. However, more studies are necessary in order to fully investigate the concordance between the phenotypic and molecular subtypes. Assessing the immunohistochemical markers from tissue microarray slides does not require the original tissue block as much as using whole sections and allows for the simultaneous analysis of large numbers of specimens. Finally, many groups have displayed excellent concordance between tissue microarray spots and whole-tissue sections [42].

To conclude, in this study, we showed that the prognostic value of serum concentrations and the tissue expressions of angiogenic growth factors VEGF, bFGF, and PDGF-bb vary according to the phenotypic subtype of the tumor. In particular, the 5-year disease-specific survival of patients with high expressions of VEGF varied remarkably between the different phenotypic subgroups. While these results are insufficient to warrant clinical implementation, it seems that a high tissue expression and serum concentration of VEGF could be used as a marker of a better prognosis in *metabolic* CRC. Moreover, as emerging studies indicate [41], drug responses to anti-angiogenic therapies may also differ according to phenotypic subtype. Additional studies are needed to further investigate these findings.

## 5. Conclusions

To conclude, in this study, we showed that the prognostic value of serum concentrations and the tissue expressions of angiogenic growth factors VEGF, bFGF, and PDGF-bb vary according to the phenotypic subtype of the tumor. In particular, the 5-year disease-specific survival of patients with high expressions of VEGF varied remarkably between the different phenotypic subgroups. While these results are insufficient to warrant clinical implementation, it seems that a high tissue expression and serum concentration of VEGF could be used as a marker of a better prognosis in *metabolic* CRC. Moreover, as emerging studies indicate (41), drug responses to anti-angiogenic therapies may also differ according to phenotypic subtype. Additional studies are needed to further investigate these findings.

## Figures and Tables

**Figure 1 cancers-15-03871-f001:**
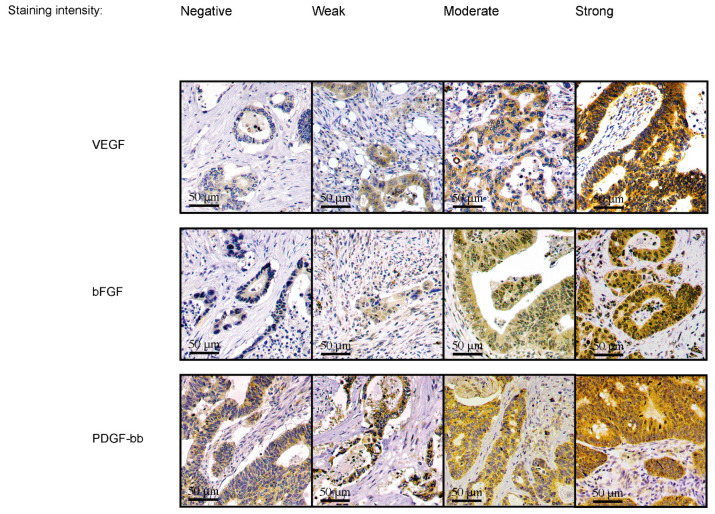
Representative images of the VEGF, bFGF, and PDGF-bb immunohistochemistry. Original magnification: ×20.

**Figure 2 cancers-15-03871-f002:**
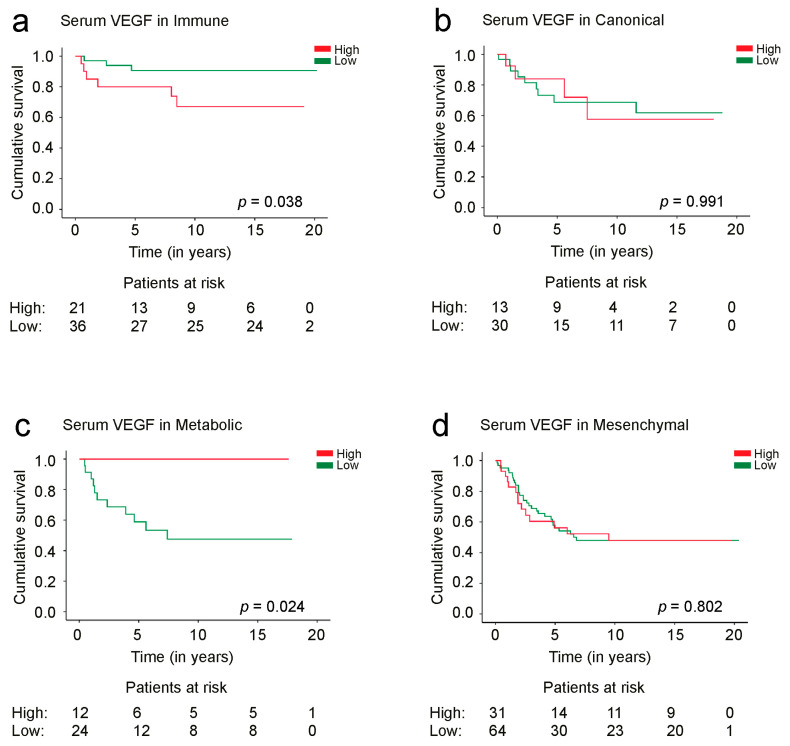
Disease-specific survival of patients with (**a**) *immune*, (**b**) *canonical*, (**c**) *metabolic*, and (**d**) *mesenchymal* tumor phenotypes according to the serum VEGF concentration. Survival curves were drawn using the Kaplan–Meier method and the *p* values are based on the log-rank test.

**Figure 3 cancers-15-03871-f003:**
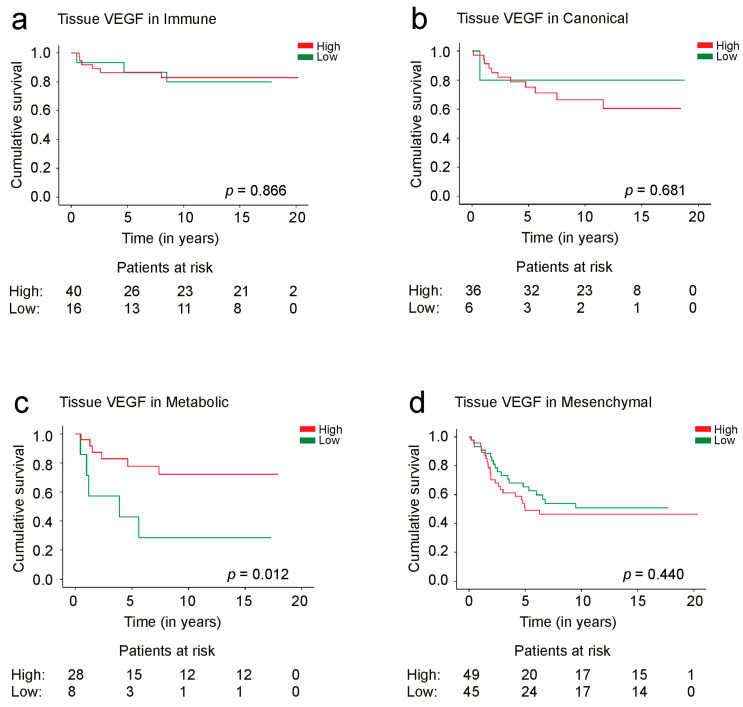
Disease-specific survival of patients with (**a**) *immune*, (**b**) *canonical*, (**c**) *metabolic*, and (**d**) *mesenchymal* tumor phenotypes according to the VEGF tissue expression. Survival curves were drawn using the Kaplan–Meier method and the *p* values are based on the log-rank test.

**Table 1 cancers-15-03871-t001:** Correlations between serum angiogenic growth factor concentrations and clinicopathological variables among 320 colorectal cancer patients.

	Serum Angiogenic Growth Factor Concentrations					
Clinicopathological Variable	VEGF		bFGF		PDGF-bb	
	**r_s_**	***p* Value**	**r_s_**	***p* Value**	**r_s_**	***p* Value**
Age	0.041	0.465	0.088	0.116	0.309	**<0.001**
Gender	0.027	0.632	0.014	0.808	0.047	0.402
Stage (I–IV)	0.039	0.494	0.053	0.347	0.039	0.490
Tumor location	0.072	0.200	0.066	0.239	0.054	0.336
Phenotypic subtype	0.088	0.181	0.036	0.583	0.117	0.076

r_s_ = Spearman’s correlation coefficient.

**Table 2 cancers-15-03871-t002:** Associations between tissue expressions of angiogenic growth factors with clinicopathological variables among 320 colorectal cancer patients.

	Tissue Expressions of Angiogenic Growth Factors								
Clinicopathological Variable	VEGF			bFGF			PDGF-bb		
Age	Low	High	*p* Value ^1^	Low	High	*p* Value ^1^	Low	High	*p* Value ^1^
≤66	51 (39.2%)	79 (60.8%)	0.466	46 (37.1%)	78 (62.9%)	0.670	33 (26.2%)	93 (73.8%)	0.772
>66	50 (35.0%)	93 (65.0%)		47 (34.6%)	89 (65.4%)		35 (24.6%)	107 (75.4%)	
Gender									
Female	56 (41.5%)	79 (58.5%)	0.129	48 (36.6%)	83 (63.4%)	0.768	36 (26.7%)	99 (73.3%)	0.624
Male	45 (32.6%)	93 (67.4%)		45 (34.9%)	84 (65.1%)		32 (24.1%)	101 (75.9%)	
Stage (I–IV)									
I	9 (23.1%)	30 (76.9%)	0.130	12 (33.3%)	24 (66.7%)	0.756	7 (18.4%)	31 (81.6%)	0.132
II	31 (35.2%)	57 (64.8%)		30 (34.1%)	58 (65.9%)		17 (19.8%)	69 (80.2%)	
III	38 (40.0%)	57 (60.0%)		31 (34.4%)	59 (65.6%)		26 (28.0%)	67 (72.0%)	
IV	22 (46.8%)	25 (53.2%)		18 (42.9%)	24 (57.1%)		17 (36.2%)	30 (63.8%)	
Tumor location									
Right colon	25 (33.3%)	50 (66.7%)	0.659	23 (31.1%)	51 (68.9%)	**0.012**	17 (23.0%)	57 (77.0%)	0.806
Left colon	56 (39.4%)	86 (60.6%)		58 (43.9%)	74 (56.1%)		38 (27.0%)	103 (73.0%)	
Rectum	20 (35.7%)	36 (64.3%)		12 (22.2%)	42 (77.8%)		13 (24.5%)	40 (75.5%)	
Phenotypic subtype									
Immune	16 (28.6%)	40 (71.4%)	**<0.001**	18 (32.7%)	37 (67.3%)	**0.022**	8 (14.5%)	47 (85.5%)	**<0.001**
Canonical	6 (14.3%)	36 (85.7%)		7 (16.7%)	35 (83.3%)		5 (11.6%)	38 (88.4%)	
Metabolic	8 (22.2%)	28 (77.8%)		15 (42.9%)	20 (57.1%)		6 (17.1%)	29 (82.9%)	
Mesenchymal	45 (47.9%)	49 (52.1%)		39 (42.9%)	52 (57.1%)		35 (38.5%)	56 (61.5%)	

^1^ Pearson’s chi-square test.

## Data Availability

All data are available upon request from the corresponding author.

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
