# Peer review of "Prognostic Values of Tissue and Serum Angiogenic Growth Factors Depend on the Phenotypic Subtypes of Colorectal Cancer"

_cancers, 2023, doi:10.3390/cancers15153871_

Round 1

Reviewer 1 Report

The authors of the manuscript “Prognostic Values of Tissue and Serum Angiogenic Growth Factors Depend on the Phenotypic Subtypes of Colorectal Cancer” for Cancerclassified colorectal cancer (CRC) patients into four phenotypic subgroups and investigated the prognostic value of several angiogenic growth factors' expression across the subgroups. 

This is an interesting work that shed some light on the prognostic value of VEGF, bFGF, and PDGF-bb expression. 

The topic has been well discussed in the introduction and the experimental project has been clearly outlined in the manuscript. Figures and tables are well crafted and they help to understand the results. 

The limitations of the study are highlighted in the discussion as the authors underlie the need for larger studies to confirm their initial results.

Minor criticisms:

-       I recommend adding other references in the introductory paragraph to further clarify the role of tissue and serum angiogenic growth factors in colorectal cancer (e.g. Gonzalez F.J. et al. J Cell Mol Med. 2007 - PMID: 17367506). 

-       In the concluding paragraph, it would be appropriate to further speculate on future developments in the field.

Author Response

Dear reviewer,

We thank you for careful reading of our manuscript and your insightful comments. Please find attached our response letter, where provide a point-by-point responses to your comments.

We feel that the revisions guided by your comments have improved our manuscript substantially and hope that you now find it suitable for publication in Cancers.  

Yours sincerely,

Jussi Kasurinen, MD

Adj. Professor Camilla Böckelman

Professor Caj Haglund

Research Programs Unit

Translational Cancer Biology

University of Helsinki

Reviewer 2 Report

The presented manuscript is concerning prognostic values of tissue and serum angiogenic Growth Factors, which depends on the phenotypic subtypes of Colorectal Cancer. The authors revealed that among the metabolic subgroup, patients with high serum concentrations of VEGF, bFGF, or PDGF-bb exhibited a significantly improved prognosis. Those patients with high VEGF tissue expressions exhibited a significantly improved prognosis among patients in the metabolic subgroup. Among immune patients, a high VEGF serum expression associated with a worse prognosis. A high serum bFGF concentration associated with a favorable prognostic factor among patients with a canonical tumor phenotype. A high PDGF-bb tissue expression associated with non metastasized disease and with the immune, canonical, and metabolic subtypes.

Paper is written in a thoughtful and understandable way. It briefly summarizes the aim of the study and is divided into individual sections in which the authors accurately explain the carried out research. In terms of content, the information was presented fairly and accurately. What is more, clear and extremely carefully made figures deserve special mention.

The presented manuscript is the valuable and interesting original article and I recommend this paper for publication in Cancers journal.

Author Response

(The authors gave the same response as above.)

Reviewer 3 Report

This study demonstrated that the expression of angiogenic growth factors in serum and in tumor tissue can be a prognostic factor in colorectal cancer patients but only when the cases are subdivided into phenotypic groups: immune, canonical, metabolic and mesenchymal. Unexpectedly, the same molecular trait, e.g. high expression of VEGF in serum can be favorable prognostic factor in one phenotypic group (metabolic) and unfavorable factor in another group (immune). The authors in the Discussion are trying to explain this apparent inconsistency. The study is well-designed, performed and reported. The overall number of patients is decent (more than 300), but when the tumors are subdivided into phenotypic groups the number of patients drops in the analyzed groups so the survival data must be interpreted with caution. Other strengths and limitations of this study are discussed in a short paragraph by the authors themselves.

Author Response

(The authors gave the same response as above.)

Reviewer 4 Report

The manuscript entitled "Prognostic Values of Tissue and Serum Angiogenic Growth Factors ..............Phenotypic Subtypes of Colorectal Cancer" is primarily focused on defining the prognostic value of Angiogenic Growth Factors (VEGF, bFGF and PDGF-bb) expression in tumor tissues and serum of CRC patients. Authors used tissues and serum collected/banked from CRC patients surgically treated approximately 20-years before between 1998 and 2003. Authors further classified these resected tumors into four consensus molecular subtypes (CMS) as defined in the Colorectal Cancer Consortium (2015), and used expression of above-mentioned angiogenic growth factors in tissues and their respective serum concentration to establish their correlation with patient's prognostic markers. As no significant correlation were observed between VEGF/bFGF and clinicopathological variables including age, gender, tumor stage/location and phenotypic subtypes (except between serum PDGF-bb concentration and age), the overall importance of this study appears to be minimal. Similarly, the tissue expression of these factors appears to be limited correlating only with tumor location and CMS1 subgroup. Similar limitation existed with the survival data. Therefore, overall importance of this study appears to be minimum, although it is well-designed and well-written.  

Although the manuscript is well written, it needs minor but thorough edits. For example, the first sentence of the 4th paragraph of Introduction appears to be incomplete. Last sentence of the 5th paragraph of introduction also appears to be incomplete.

Author Response

(The authors gave the same response as above.)

Round 2

Reviewer 4 Report

Thanks for responding to comments and concerns raised by the reviewers.